# Learning to Communicate Through Implicit Communication Channels

**Han Wang**
The Chinese University of Hong Kong, Shenzhen
xwanghan@gmail.com

**Binbin Chen**
ByteDance Inc.
chenbinbin.1996@bytedance.com

**Tieying Zhang**
ByteDance Inc.
tieying.zhang@bytedance.com

**Baoxiang Wang**[*]
The Chinese University of Hong Kong, Shenzhen
Vector Institute
bxiangwang@cuhk.edu.cn

## Abstract

Effective communication is an essential component in collaborative multi-agent systems. Situations where explicit messaging is not feasible have been common in human society throughout history, which motivate the study of implicit communication. Previous works on learning implicit communication mostly rely on theory of mind (ToM), where agents infer the mental states and intentions of others by interpreting their actions. However, ToM-based methods become less effective in making accurate inferences in complex tasks. In this work, we propose the Implicit Channel Protocol (ICP) framework, which allows agents to communicate through implicit communication channels similar to the explicit ones. ICP leverages a subset of actions, denoted as the scouting actions, and a mapping between information and these scouting actions that encodes and decodes the messages. We propose training algorithms for agents to message and act, including learning with a randomly initialized information map and with a delayed information map. The efficacy of ICP has been tested on the tasks of Guessing Numbers, Revealing Goals, and Hanabi, where ICP significantly outperforms baseline methods through more efficient information transmission.

## 1 Introduction

Effective communication is pivotal in collaborative multi-agent systems, especially in environments characterized by incomplete information (Panait & Luke, 2005; Busoniu et al., 2008; Tuyls & Weiss, 2012). Communication acts as a vital conduit, enabling agents to exchange private information, co-ordinate joint actions, and infer real-world states (Wang et al., 2021). These processes synergistically foster tighter cooperation and enhance collective performance (Li et al., 2002; Cao et al., 2012). We focus on multi-agent reinforcement learning (MARL) methods for communication, where communication is broadly categorized into explicit and implicit strategies (Dafoe et al., 2020).

Explicit communication uses direct channels independent of the environment dynamics (Sukhbaatar et al., 2016; Foerster et al., 2016; Jiang & Lu, 2018), allowing agents to transmit observations, intentions, and advice to facilitate decision-making and coordination (Zhu et al., 2022; Qu et al., 2021). This approach, analogous to human language or verbal exchanges (Havrylov & Titov, 2017; Baker et al., 1999), has been widely employed in MARL to enhance collaboration. However, dependence on direct channels introduces significant computational and memory overheads (Roth et al., 2006),

---

[*]Corresponding to: Baoxiang Wang

which makes it challenging to implement in certain scenarios. Examples are tasks without communication channels or decentralized frameworks (Oliehoek et al., 2008; Kraemer & Banerjee, 2016).

Situations where explicit messaging is not feasible have been common in human society throughout history. From early humans engaging in hunting and gathering through silent cooperation (Klein, 2009; Tomasello & Vaish, 2013), to modern military operations using gestures and codes for covert communication (Tzu, 2008), and even in everyday social interactions where intentions are conveyed through expressions, tone, and body language (Pease, 1984; Duncan Jr, 1969). Implicit communication has established an effective mechanism for information sharing without explicit language.

Learning methods for implicit communication have been investigated by the MARL community. A prominent method is the theory of mind (ToM) (Premack & Woodruff, 1978), where agents infer the mental states and intentions of others by interpreting their actions (Heider & Simmel, 1944). By modeling the beliefs, desires, and intentions of other agents, ToM enables agents to coordinate in a variety of simple tasks (Baker et al., 2017; Zhao et al., 2023; Nguyen et al., 2020). However, ToM-based approaches face significant challenges, including the difficulty of making accurate inferences and the high computational complexity involved in modeling other agents. These issues become particularly pronounced in dynamic environments where agents must constantly update their models based on limited or ambiguous information.

To address these challenges associated with ToM methods, we introduce Implicit Channel Protocol (ICP), a novel framework that allows agents to communicate through communication protocols in implicit communication likewise how it was done in explicit communication. ICP leverages a subset of actions, denoted as the scouting actions, which have no or uniform effects on environment dynamics. A centralized mapping between information and these scouting actions is established to encode and decode the messages. Agents exchange information by deliberately taking scouting actions, forming an implicit communication channel. We further demonstrate how agents' strategies are trained on this channel, including training with a randomly initialized information map and training with a delayed information map.

We validate the effectiveness of ICP through comprehensive experiments on the tasks of *Guessing Numbers*, *Revealing Goals*, and *Hanabi* (Bard et al., 2020). These environments share a common characteristic: they lack direct communication but agents must collaboratively make decisions to achieve shared rewards. This setting introduces significant challenges, including sparse and delayed reward feedback, along with difficulty in credit assignment both temporally and among agents. Despite these hurdles, our experiments on *Guessing Numbers* and *Revealing Goals* demonstrate that ICP significantly enhances performance, through more efficient information transmission, compared to baseline methods. In *Hanabi*, which is a popular card game played by humans, our approach achieved an average score of 24.91 out of 25, which surpasses the best available learning algorithm which obtains 23.81.

## 2 BACKGROUND AND RELATED WORK

**Decentralized Partially Observable Markov Decision Process** The cooperative multi-agent problem can be formulated as a Dec-POMDP game (Bernstein et al., 2002), which is described by the tuple $G = (S, U, N, \mathcal{T}, O, R, \gamma)$, where $S$ represents the true global state of the environment. At each discrete time step $t$, every agent $i \in N := \{1, \ldots, n\}$ selects an action $u_i$ from its action space $U_i$. The state transition function $\mathcal{T}(s'|s, u) : S \times U \times S \to P(S)$ governs the transition of states, where $u = (u_1, \ldots, u_n)$ denotes the joint action. In POMDP, the global state remains inaccessible, and each agent $i$ can only perceive its individual observation $o_i$ through the observation function $O_i(s) : S \times N \to O$. $R_i(s, u_i) : S \times U_i \to R$ represents the reward function for each agent $i$. In the cooperative scenarios, all agents share a common reward function $R(s, u) : S \times U \to R$, known as team reward. The objective for each agent is to maximize the expected return, which makes effective cooperation among all agents necessary.

**Learning to Communicate** Communication among agents is critical for effective collaboration in MARL. Most researches in this field focused on explicit communication protocols (Tucker et al., 2022; Peng et al., 2017; Kong et al., 2017; Pesce & Montana, 2020; Kim et al., 2019; Wang et al., 2020; Freed et al., 2020b;a; Gupta et al., 2023; Foerster et al., 2016), where agents exchange messages containing critical information. Additionally, techniques like attention mechanisms or graph-

based methods that used to build more effective communication connection (Jiang & Lu, 2018; Das et al., 2019; Jiang et al., 2018; Sukhbaatar et al., 2016; Chen et al., 2024; Tucker et al., 2022) and intention sharing or theory of mind (ToM) reasoning to transmit more useful information (Wang et al., 2021; Kim et al., 2020; Qu et al., 2021), further enhancing the effectiveness of MARL systems.

Explicit communication on direct channels often incurs significant communication overhead and may not be available in environments with limited bandwidth and high communication costs. Implicit communication, where agents convey information through their behaviors, has been explored as an alternative approach (Li et al., 2024; Tian et al., 2023; Li et al., 2021; Shaw et al., 2022; Grupen et al., 2022). ToM-based methods represent one main research strand for implicit communication, where agents infer the intentions and beliefs of others to anticipate actions and adjust strategies accordingly. Methods in Rabinowitz et al. (2018); Tian et al. (2020); Nguyen et al. (2020); Zhao et al. (2023) aim to enhance coordination by modeling and predicting others' behavior based on observed actions. However, ToM-based approaches can be computationally intensive and may struggle to accurately infer intentions in complex environments.

## 3 SETTING

Consider in a fully cooperative Dec-POMDPs, where no direct communication channel is available. Each agent maintains a joint action-observation history $\tau_{t,i} = \{o_{0,i}, u_0, r_1, ..., r_t, o_{t,i}\}$ and makes decisions and executes actions based on it. Within this setting, we define a subset of actions as *scouting actions* $U^s$ which have no or uniform effects on the state $s_t$ or reward $r_t$ and mainly affect the observation function $O_i(s_t)$. The remaining actions are defined as *regular actions* $U^r$.

In an idealized situation, scouting actions serve to influence the observation mapping function, which allows the agents to gather critical information from the environment to improve their decision-making and coordination. A scouting action might reveal information about an agent's surroundings or the state of other agents without changing the environment unpredictably. For instance, in the game of *Hanabi*, *Hint* actions reveal information to other players while consuming an information token, thereby uniformly altering the game state. Similarly, in *StarCraft II*, scouting the map with *Scan* from *Terrans* requires depleting energy, which impacts the environment and provides new information in the agent's observations.

We are interested in this setting because agents generally need to employ scouting actions to collect sufficient information before taking other actions to obtain rewards. Both the information collection phase and the reward acquisition phase may require cooperation among agents. For example, agents will only receive positive rewards if one agent scouts the correct information for another agent, and the latter correctly utilizes this information. In this context, sparse and delayed reward feedback, along with the credit assignment problem both temporally and among agents, poses significant challenges for agents to learn optimal strategies.

## 4 INFORMATION CARRIED BY SCOUTING ACTIONS

Scouting actions carry two types of information.

1. **Information reflected through the environment.** When an agent performs a scouting action, it influences the observation function, and the new observations provide information. This type of information depends on the observation function, the states, and the joint actions of all agents.

2. **Information reflected through the choice of the scouting action.** The specific scouting action chosen by an agent can carry intentional meaning. This type of information depends only on other agents' scouting action policy. If the receiver agent cannot understand the sender agent's intention, no information will be transmitted.

General multi-agent reinforcement learning (MARL) methods tend to focus more on the first type of information. In Dec-POMDPs without modeling other agents' strategies, both partial observation of states and other agents' policies introduce uncertainty. In the exploration phase of online MARL training, the second type of information cannot be utilized due to the unknown policies of others. Therefore, MARL methods mostly learn to use the information reflected through the environment.

For the second type of information which is not limited by observation function and states, when agents' intention is understood correctly, it is more fixable and could be more useful for decision-making and cooperation. Theory of mind (ToM) methods utilize the second type of information by modeling strategies or inferring the intention of other agents. By estimating the internal states of other agents, an agent can interpret the intentional choices behind their scouting actions. One approach is to build an explicit belief model that infers the state from observed actions to achieve communication. However, this method incurs a computational cost that grows exponentially with the size of the game and the number of agents. Moreover, the precise state distribution information is often difficult to fully utilize. Another approach involves training a neural network to map actions to states or intentions, which requires exponentially more data compared to action policy training. Since this method does not produce explicit inferences, the resulting information is inherently biased, which could negatively impact the agent's decision-making process.

We propose a simpler and more effective solution to take advantage of the second type of information conveyed by scouting actions while also mitigating the computational overhead and inaccuracies of ToM methods. By establishing a shared communication protocol where all agents follow the same strategy for selecting scouting actions, the intended information can be easily decoded from the chosen action. This approach eliminates the need for state estimation and intention inference required in ToM methods. In our setting, this communication protocol has the following benefits.

1. **Low Computational Overhead and Accurate Information Transmission.** By having all agents follow the same strategy for selecting scouting actions, there is no need for additional computational inference or training of a separate inference model to deduce intentions. The inverse of the strategy can be used to directly interpret the information, resulting in low computational overhead and accurate transmission of intentions.

2. **Efficient Communication Enabled by Broadcasted Information.** Scouting actions are observable by all agents in the system, allowing any information conveyed through these actions to be inherently broadcasted. This characteristic facilitates the use of efficient embedding techniques, such as the hat mapping method, to achieve more effective communication.

3. **Independence from Observation-Based Information.** When different embeddings are used, the information conveyed by altering the observation function may differ, making the information carried by scouting actions not necessarily correlated with the environment-based information. Both types of information can be optimized concurrently to enhance cooperation and decision-making.

## 5 STRATEGY TRAINING ON CONSTRUCTED IMPLICIT CHANNEL

To establish a shared communication protocol, we propose Implicit Channel Protocol (ICP), a new framework for agents' implicit communication. In this framework, at each step, agents choose whether to send information. If an agent chooses not to send information, a regular action $u^r$ will be taken according to the action policy. If otherwise an agent wishes to convey message $m_i \in M$, it will output a scouting action $u^s = \mathcal{P}(m_i)$, which maps $m_i$ through the centralized mapping mechanism $\mathcal{P}$. This mechanism $\mathcal{P}$ needs to be decodable, which provides output messages from scouting actions. By allowing agents to send information through encoded scouting actions and other agents to decode the messages from the globally observed scouting actions, an implicit communication channel is constructed.

As agents need to decide whether to send information, the new action space becomes $U' = \{U - U^s, send\_info\}$, then agents sample actions $u_i \in U'$ from new action policy $u_i \sim \pi_i(\cdot \mid \tau_{t,i}, m_{-i})$, where $m_{-i}$ are decoded messages transmitted from other agents. For message $m_i$, it is selected by the message strategy $m_i \sim \phi_i(\cdot \mid \tau_{t,i}, m_{-i})$. Based on this process, we can formulate agents' value functions as $V^{\pi_i,\phi_i,\mathcal{P}}(s) := \mathbb{E}[\sum_{t=1}^{T} \gamma^{t-1} r_t(s_t, u_t, s_{t+1}) \mid s_1 = s]$, for $s_{t+1} \sim \mathcal{T}(\cdot \mid s_t, u_t)$ and $u_t = (u_{t,1}, ..., u_{t,n})$,

$$\text{and } u_{t,i} = \begin{cases} u^s = \mathcal{P}(m_i \sim \phi_i(\cdot \mid \tau_{t,i}, m_{-i})) & \text{if } send\_info, \\ u^r \sim \pi_i(\cdot \mid \tau_{t,i}, m_{-i}) & \text{else.} \end{cases} \tag{1}$$

Equation (1) highlights three essential components of the framework: the agents' action policy $\pi$, the message strategy $\phi$, and the centralized mapping mechanism $\mathcal{P}$. To develop these components, we investigate two different methodologies. The first one is training with a random initial information map and the second one is training with a delayed information map. Both approaches construct an implicit communication channel within the environment, with which the agents exchange information and achieve implicit coordination. In the rest of the section, we will discuss the details of these approaches.

## 5.1 STRATEGY TRAINING WITH RANDOM INITIAL MAP

In this approach, we first establish a one-to-one mapping between information and actions using a randomly initialized embedding, that makes the size of the message the same as the size of the scouting action space $M = \{1, ..., |U^s|\}$. Once this one-to-one mapping is set, the inverse of $\mathcal{P}$ will obtain the original message $m_i = \mathcal{P}^{-1}(u_i^s)$.

In our implementation, we use a value-based method for training the action policy, incorporating parameter sharing and value decomposition techniques to facilitate effective coordination among agents. For the information strategy, we use the communication gradient method (Foerster et al., 2016), which is designed to optimize communication protocols within a predefined channel. Both the action policy and the information strategy are parameterized as Q-networks, for action policy $\pi_i(\tau_{t,i}) = \arg\max Q^{\theta_{1,i}}(\tau_{t,i}, u_i)$, and message strategy $\phi_i(\tau_{t,i}) = \arg\max Q^{\theta_{2,i}}(\tau_{t,i}, m_i)$. Given the value function $V^{\pi_i,\phi_i,\mathcal{P}}(s)$, the Q-functions are defined by $Q^{\pi_i}(\tau_{t,i}, u_i) := \mathbb{E}_{m \sim \phi, \mathcal{P}}[r_t(s_t, u_t, s_{t+1}) + \gamma V^{\pi_i,\phi_i,\mathcal{P}}]$, and $Q^{\phi_i}(\tau_{t,i}, m_i) := \mathbb{E}_{u^r \sim \pi, \mathcal{P}}[r_t(s_t, u_t, s_{t+1}) + \gamma V^{\pi_i,\phi_i,\mathcal{P}}]$, where $\pi = (\pi_1, ..., \pi_n)$, $\phi = (\phi_1, ..., \phi_n)$ and $m = (m_1, ..., m_n)$.

**Parameter Sharing and Value Decomposition** Parameter sharing enables different agents with the same observation and action spaces to learn from a shared network. Despite using the same network, agents evolve different hidden states and receive distinct observations, allowing them to behave differently. We incorporate the agent index $i$ into the network input, allowing for specialization through rich representations in deep Q-networks.

To further enhance learning efficiency, we apply value decomposition (Sunehag et al., 2018) for the action policies $\pi$. The joint action-value function $Q_{\text{tot}}^\pi(\tau, u)$ is expressed as the sum of individual value functions: $Q_{\text{tot}}^\pi(\tau, u) = \sum_{i=1}^N Q^\pi(\tau_{t,i}, u_i; \theta_{1,i})$, where $\tau = (\tau_i, ..., \tau_N)$. This decomposition simplifies learning by allowing the gradients to propagate through individual agents' policies while maintaining a shared reward structure, which accelerates the overall training process.

**Communication Gradient** Since our message space is $M = \{1, ..., |U^s|\}$, which is discrete and non-binary, we introduce the Gumbel-Softmax technique (Havrylov & Titov, 2017; Jang et al., 2016) to sample one-hot vector messages $\vec{m}$. The Gumbel-Softmax method enables us to generate discrete samples that are differentiable, facilitating end-to-end training with gradient descent. Specifically, we sample the one-hot vector message $\vec{m}$ such that $\vec{m}_i = 1$ if $i = \arg\max_{j \in M}(\log(\phi(\cdot \mid \tau_{t,i}, m_{-i}) + g_j))$, and $\vec{m}_i = 0$ otherwise, where $g_j = -\log(-\log(u_j))$ and $u_j \sim U(0, 1)$.

This method ensures a continuous relaxation between $\phi$ and $\vec{m}$. As a result of this relaxation, the game becomes fully differentiable and can be trained using the backpropagation algorithm. By enabling gradients to flow smoothly through the communication process, this approach facilitates more effective and efficient training of agents. Comparing the communication gradient implementation in DIAL (Foerster et al., 2016), we adopt this Gumbel-Softmax technique to replace the Discretize/Regularize Unit (DRU). This substitution allows agents to send discrete and non-binary messages during the learning phase, thereby enhancing the richness and accuracy of the information exchanged.

## 5.2 STRATEGY TRAINING WITH DELAYED MAP

In the training process with a delayed map, we begin by learning the information strategies and action policies using a direct communication channel. This initial phase provides agents with a flexible environment that allows for direct communication and enable them to develop their strategies more

effectively. Once these strategies are established, we determine the mapping mechanism that maps the messages to actions. Finally, we fine-tune both the action policies and information strategies in the ICP framework to ensure optimal performance within the constraints of the final environment.

Given that the information strategies were initially trained using a direct communication channel, there may be discrepancies between this channel and the implicit communication channel constructed by scouting actions, particularly in terms of capacity and whether the channel is discrete or noiseless. To minimize any potential performance loss resulting from these differences, we can limit the capacity of the direct channel or apply efficient embedding techniques (such as the hat mapping method) during the message-to-action mapping process. These steps help to align information strategies developed in the direct channel with the constraints of the implicit channel and ensure messages are decodable.

During implementation, we utilize standard explicit communication algorithms designed for direct channels, such as RGMComm, to pre-train the information strategy. Then we use two kinds of mapping methods, including simple one-to-one mapping and the hat mapping method (Brown & Tanton, 2009; Bushi, 2012; Butler et al., 2009; Feige, 2004; Havil, 2011; Winkler, 2002) to obtain new information strategies. The one-to-one mapping ensures that each message corresponds directly to an action, while the hat mapping allows an agent to communicate efficiently with multiple receivers. Finally, we fine-tune both the information strategy and the action policy to optimize performance within the implicit communication framework.

**Pre-training with Direct Channel**    In the pre-training phase, we utilize a direct communication channel to facilitate the learning of the communication strategies. During this phase, the message size can be larger than the scouting action space, allowing for more complex and detailed communication between agents. This flexibility enables the agents to explore a wider range of communication strategies and coordination behaviors, which are crucial for solving complex tasks in partially observable environments. However, after pre-training, the message strategy needs to transition to the implicit communication framework and it is necessary to compress these messages without loss of information. For one-to-one mapping, this mapping requires that the message size matches the scouting action space exactly. It ensures a direct and consistent mapping between the learned messages and the corresponding actions.

**Hat Mapping Implementation**    When the local observations of different agents' have large overlaps, the hat principle-based mapping method utilizes broadcast channels to achieve a single transmission of messages to multiple receivers. The method is inspired by the multi-color hat guessing game, where players use logical deduction based on limited communication to maximize team success. In this game, players can observe the hat colors of others but not their own, and they must guess their own hat color using a pre-defined strategy. A well-known strategy involves each player leveraging the sum of the hat colors they observe, modulo the total number of possible colors. For example, in an 8-color version of the game, each player can compute the sum of the colors they observe (assigned numerical values) and use a pre-determined rule, such as calculating the sum modulo 8, to make a logical deduction about their own hat color.

In our implementation, at each step, each agent computes local messages intended for every other agent. These local messages are derived from the shared information strategy and the observations that the target agent cannot see but are overlapped with other agents—much like the "hat" in the hat guessing game. When an agent sends information, it transmits a structured public message that combines all these local messages using a sum modulo operation. As a result, each receiving agent can also use the sum modulo operation to combine the received public message with the local messages of all agents except themselves and the sender. This allows them to infer their own specific local message from the combined public message. This method enables a single broadcast to convey individualized information to multiple agents simultaneously, enhancing communication efficiency even when explicit communication is not possible.

## 6    EXPERIMENTS

In this section, we explore the application of the ICP across various environments where no direct communication is available. ICP plays a vital role in these settings by constructing an implicit

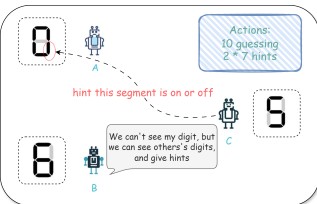 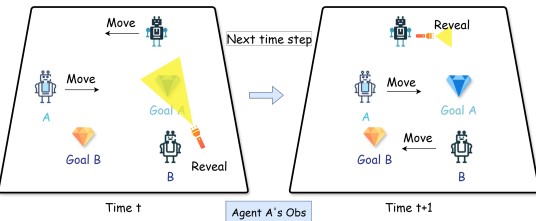

Figure 1: **Left**: *Guessing Numbers Environment*. Agents cannot see their own digits, but can reveal others' segments' states by collaboratively giving hints. They can deduce their own digit and obtain shared rewards. **Right**: *Revealing Goals Environment*. The agents are positioned in a random spots in a grid world and are assigned a unique target. However, they can only observe others' targets which do not include themselves. By revealing each other's targets in the nearby grid, they can eventually find their own targets and reach them.

communication channel that enhances the agents' ability to share and interpret information. We will demonstrate how ICP facilitates effective collaboration and decision-making in 3 tasks, namely *Guessing Numbers*, the *Revealing Goals*, and the *Hanabi* card game. The first two tasks are designed by us and can be reused by future works as testing environments. The Hanabi game is a popular card game played by humans, and is described as *The Hanabi Challenge* (Bard et al., 2020) by the community. Each of these environments presents unique challenges, and we will show how ICP optimizes the use of scouting actions to improve overall performance and strategy.

## 6.1 GUESSING NUMBERS

In the *Guessing Numbers game*, we present a collaborative, turn-based multi-agent reinforcement learning environment involving $N$ agents. Each agent $i$ is uniquely assigned a digit $d_i \in \{0, 1, \ldots, 9\}$, which is visible to all other agents $j \neq i$ but remains unknown to agent $i$ themselves. The primary objective for each agent is to deduce and correctly guess their own digit $d_i$. During each turn, an agent can choose to either *guess* their own digit by selecting $\hat{d}_i \in \{0, 1, \ldots, 9\}$ or provide a *hint* about another agent's digit using the *Radioland Slim* font—a seven-segment digital display representation. Specifically, hints involve revealing the state ("on" or "off") of one of the seven segments of another agent's digit, and these hints are public and observable by all agents. Each agent's action space comprises $10 + 7 \times (N - 1)$ actions: 10 options for guessing their own digit and 7 hint options for each of the $N - 1$ other agents. The game encourages strategic collaboration, as agents must balance between gathering information to deduce their own digit and assisting others through informative hints, aiming for the collective success of all agents correctly guessing their digits.

In this game, successfully guessing their digit rewards an agent with 10, while performing a hint action incurs a small penalty of $-0.1$ to encourage efficient communication and collaboration. The game imposes a limit $l$ on the number of hint actions, and each agent is allowed only 1 guess. The game concludes when all agents have made their guesses, with the goal being to accurately guess all digits within these constraints.

In the *Guessing Numbers* experiment, we evaluate the performance of 5 approaches: VDN-on-policy, VDN-off-policy, ICP with the random initial map approach (ICN-DIAL-RM), ICP with the delayed map approach (ICN-DIAL-DM), and a cheating approach where a direct communication channel is available (DIAL-Cheat). Each approach is evaluated over 1k episodes with 6 random seeds, and running on a Linux metal machine with 256 GB RAM and 3090Ti GPU for 36 hours.

For VDN-off-policy, we begin by warming up the replay buffer until its size exceeds the batch size. During each training step, we add 10 episodes to the replay buffer and randomly sample a batch of episodes from the buffer for training. In contrast, for VDN-on-policy and our proposed method, we utilize a vectorized environment to sample a batch of episodes at each training step and use these samples for training.

Despite differences in network architecture, these approaches share similar hyperparameters. Specifically, we set the hidden size of the MLP and GRU to 256, use 2 layers in the GRU, and set the

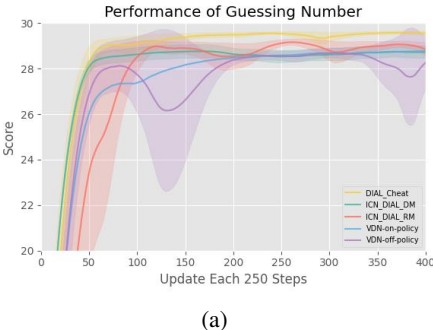 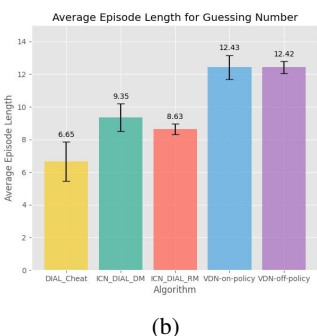

(a) (b)

Figure 2: **(a)**: The training curves of *Guessing Numbers* over a total of 100k steps with $N = 3, l = 11$. **(b)**: Average episode length running by different algorithms in *Guessing Numbers*.

learning rate to $5 \times 10^{-4}$, batch size to 256. The target network update rate is set to 10, $\gamma$ is set to 0.99, $\epsilon$ is set to 0.1, and we apply gradient clipping with a threshold of 10.

The results are presented in Figure 2. All algorithms are capable of eventually guessing their own digit correctly, but the differences of number of hit actions needed is significant among the algorithms. By examining the saved models and averaging over 10k episodes, we find that **ICN-DIAL-RM has an average episode length of approximately 8.63, while VDN-on-policy and VDN-off-policy have average episode lengths of around 12.4**. This observation further demonstrates that ICN achieves more efficient information transfer by using scouting actions in implicit channels.

## 6.2 REVEALING GOALS

In the *Revealing Goals task*, we present a non-sequential collaborative that emphasizes information sharing among agents. The environment consists of $N$ agents operating within an $H \times H$ grid world. Each agent $i$ starts at a random position and is assigned a unique goal location $g_i$ that is at least two grid units away from its starting position. Critically, agents cannot perceive their own goal locations; they can only observe the goal locations of other agents $j \neq i$. At each time step, agents select from a fixed action space of eight actions: moving *up*, *down*, *left*, or *right*, and *reveal* information about the adjacent grid cells in these directions. When an agent performs a *reveal* action, the adjacent grid cell in the specified direction becomes *revealed*, and all agents gain information about whether their own goals are located in that cell. The grid world features wrap-around edges, creating a toroidal topology where, for example, moving left from cell $(0, y)$ leads to cell $(H - 1, y)$.

An agent observes the locations of all agents, the goal points of other agents, and any goals on the revealed grids. Since they cannot directly perceive their own goals, agents must rely on information revealed by themselves and others to infer the locations of their own goals. The objective is for each agent to navigate to its designated goal location $g_i$, upon which all agents receive a shared reward of 1. When an agent reaches its goal, a new goal is randomly assigned to it but is at least two grid units away from its current position. The game proceeds for a total of $T$ time steps. This environment encourages strategic collaboration, as agents must balance between exploring the grid to find their own goals and assisting others by revealing grid cells that may contain teammates' goals. The agents will need to communicate implicitly, using the fixed action space of size 8.

In the experiments for the *Revealing Goals* task, we use the same training hyperparameters and model architecture as in the *Guessing Numbers* experiments. We also compare the performance of 5 approaches: DIAL-Cheat, ICN-DIAL-DM, ICN-DIAL-RM, VDN-on-policy, and VDN-off-policy. The experimental results are shown in Figure 3(a). **ICN-DIAL-RM significantly outperforms the baseline methods by achieving an average score 2.17 times that of the baselines**. One reason for its effectiveness is that, in each round, each of the $N$ agents can engage in $N$ broadcast communications through implicit channels, which significantly enhances information efficiency compared to relying solely on environmental feedback. This result further validates the effectiveness of ICP in more complicated grid world tasks.

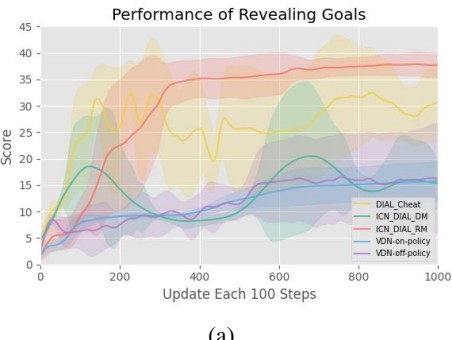
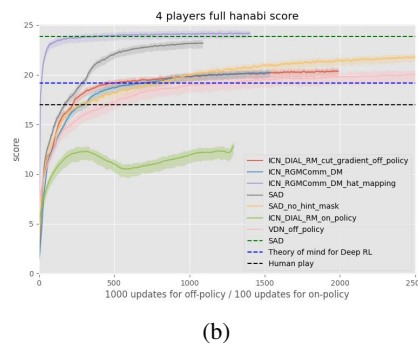

(a)                          (b)

Figure 3: **(a)**: The training curves of *Revealing Goals* over total 100k train steps with $N = 4, H = 5, T = 50$. **(b)**: The training curves of 4-players Hanabi with on-policy algorithms take around **150 hours** and off-policy algorithms take around **20 hours**.

## 6.3 HANABI

To demonstrate ICP's broader applicability, we also evaluate it in the game of *Hanabi*. *Hanabi* is a fully cooperative card game where players work together to play cards in a specific order to form sequences, aiming to achieve the highest possible score. When it is played by humans, people will mute themselves and coordinate only through taking and observing actions. This game requires effective implicit communication and strategic information sharing among the agents.

Because the game rule of Hanabi is quite involved, we refer it to a popular website `https://hanabi.github.io/` that discusses both rules and human-play strategies. The action space for each agent includes three options: playing a card out of their 5-card hand, discarding a card, or giving a hint towards another player's 5-card hand. The hint could reveal all cards in a player's hand of a specific color or rank. Since they cannot see their own cards, agents must rely on hints from others to infer what cards they hold. The success of the game heavily depends on the agents' capacity to use these hints effectively to coordinate their actions and play the correct cards in order.

In our experiments, we focus on the 4-player full version of *Hanabi*. Since hint actions in *Hanabi* are natural scouting actions, we select them to serve as information carriers for our ICP implementation. We implement several baseline algorithms, including VDN-off-policy and the original version of SAD (Hu & Foerster, 2019), as well as a variant of SAD with the hint action masks removed. For the ICP algorithm, we explore several variations, including ICN-DIAL-RM-on-policy, which involves strategy training with a random initial map using DIAL's communication gradient technique resulting in an on-policy learning method. We also implement ICN-DIAL-RM-cut-gradient-off-policy, similar to ICN-DIAL-RM-on-policy but with the communication gradient chain cut off to enable off-policy learning. Additionally, we use ICN-RGMComm-DM, which is strategy training with a delayed map using RGMComm (Chen et al., 2024), an off-policy MARL communication algorithm, and ICN-RGMComm-DM-hat-mapping, an extension of ICN-RGMComm-DM that utilizes the hat mapping method for improved communication.

Our presented results are based on the best runs selected from more than three random seeds. From the results shown in Figure 3(b), we observe that the ICP method using the hat mapping approach, *ICN-RGMComm-DM-hat-mapping*, achieved the best performance. **The score achieved by ICP is 24.91, which surpasses the best available learning algorithm (SAD, 23.81 points) and vastly outperforms theory of mind-based methods (Fuchs et al. (2021) 19.13 points) and average human players (17 points, reported in Kantack (2021)).** It is worth remarking that the state-of-the-art strategy in the *Hanabi* game is a search-based algorithm, WTFWThat+search, which reports an averaged score of $24.96$ points (Lerer et al., 2020). With the maximum score of the game being 25, and there are certain deals that prevent the players from winning 25 points in any way, both ICP and WTFWThat+search are close to fully solving the game.

For additional details on our implementation, please refer to Appendix A. A comparison between ICP and Hanabi human conventions is provided in Appendix B, along with a detailed analysis of the Hanabi results in Appendix C.

# 7 DISCUSSIONS

## 7.1 COMPATIBILITY OF ICP WITH OTHER TRAINING METHODS

In the ICP framework, action policy allows for the use of various algorithms for training. For instance, the action policy can be trained using value-based methods like VDN (Sunehag et al., 2018) or policy-based methods such as MADDPG (Lowe et al., 2017). This compatibility ensures that ICP can be tailored to various problem settings and agent dynamics. One could choose the most suitable algorithm depending on the specific characteristics of the environment and the agents.

The communication strategy within the ICP framework is also versatile. It can be implemented using a variety of direct channel communication algorithms that support discrete channels. These include well-established methods like DIAL (Foerster et al., 2016) and RGMComm (Chen et al., 2024), which are designed to optimize communication under different constraints and requirements. The flexibility in choosing both the action policy and the communication algorithm highlights the comparability and broad applicability of the ICP framework, making it a general approach for multi-agent reinforcement learning scenarios.

## 7.2 POTENTIAL OF FURTHER UTILIZING THE ENVIRONMENT INFORMATION

As described in Section 4, scouting actions carry two types of information. The ICP framework utilizes the information reflected through the choice of the scouting action (choice information) to establish an implicit communication channel. However, even when we leverage this information, the information reflected through the environment (environment information) remains useful. Although in most cases the constructed implicit channel transmits more stable and useful information, in some situations the environment information is also significant. For example, in the *Revealing Goals* environment, a revealing action intended to transmit embedded information might also reveal other agents' goals, which helps agents to make better decisions.

To verify the impact of this environmental information on performance, we randomly shuffle 6 different embeddings on the ICN-RGMComm-hat-mapping policy saved from the previous *Hanabi* experiment. We then fine-tune the action policy while keeping the information strategy fixed. The results are shown in Figure 4 in the appendix, where, under a fixed information strategy, some randomly shuffled embeddings enjoy improved performance. This suggests that one could further utilize the environment information for more effective communication in future works.

# 8 CONCLUSION AND FUTURE WORK

The Implicit Channel Protocol (ICP) introduced in this paper is an advancement in implicit communication through multi-agent reinforcement learning (MARL). By mapping information to scouting actions to construct implicit channels and optimizing both action and communication strategies, ICP facilitates communication protocols to be established even without explicit channels. The experimental results validate the protocol's effectiveness across various MARL scenarios and demonstrate its capacity to significantly improve both coordination among agents and performance in the tasks.

A possible direction for future work is extending ICP to environments that lack inherent scouting actions. In such settings, identifying appropriate actions to serve as information carriers becomes challenging, as these actions must both convey information effectively and interact with the environment beneficially. Future research could focus on developing algorithms that dynamically identify or design such actions, balancing effective communication with minimal impact on the environment. These advancements would enhance the applicability of ICP across a wider range of scenarios, making it a more versatile tool in multi-agent communication.

REPRODUCIBILITY STATEMENT

The code and our designed environments are freely available in the supplementary material. The details of the implementation can be found in Appendix A. The hyper-parameters used in experiments are also mentioned in Section 6.

ACKNOWLEDGEMENT

Han Wang and Baoxiang Wang are partially supported by the National Natural Science Foundation of China (62106213, 72394361), an extended support project from the Shenzhen Science and Technology Program, Longgang District Key Laboratory of Intelligent Digital Economy Security, and ByteDance.

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

# A  TRAINING DETAILS

## A.1  ICP IMPLEMENTATION WITH DIAL AND VDN

In our implementation, we employ a 2-head input and 2-head output Implicit Channel Net (ICN) to parameterize all Q functions. Specifically, at each time step $t$, for agent $i$, observation-action pairs $(o_{t,i}, u_{t-1})$ are input to one Multilayer Perceptron (MLP) and Rectified Linear Unit (ReLU) layer. Additionally, the last message vectors from other agents ($\vec{m}_{k,l}$, where $k = \max(j \mid \vec{m}_{j,l} \in \vec{M})$) for $l \in (1, \ldots, N)$ are stacked together along with each message sender's ID, then input to another MLP and ReLU layer. The outputs of these two layers are summed and input to a Gated Recurrent Unit (GRU) network (Chung et al., 2014). This process approximately achieves the Q function input with action-observation history $\tau$ and other agents' messages $m_{-i}$ by utilizing hidden states and inputs at each time step.

Subsequently, the output of the GRU is directed to two distinct heads: an action head and a message head. The action head generates logits for actions, with an output dimension of $|U - U^s| + 1$, while the message head is specialized to handle the agent's communication behavior, producing logits for messages. Specifically, the output dimension of the message head is set to $|U^s| + 1$, where the additional dimension represents the *NOOP* (no operation) action. This design ensures that the agent refrains from sending a message when the action sampled from the action logits does not correspond to *send_info*. To enforce this behavior, a mask is applied to the output of the message head. When the action sampled from the action logits is *send_info*, the mask restricts Gumbel-Softmax sampling to select the *NOOP* action. If the agent decides to send a message, the sampled one-hot vector of messages $\vec{m}$ directly maps to the $j$-th information action $u^{\text{info}}$, where $j = \text{argmax}_{l \in (1, \ldots, |U^s|)} \vec{m}_l$.

During centralized training sessions, action sampling follows an $\epsilon$-greedy policy to ensure exploration, while we maintain the gradient of the one-hot vector of messages $\vec{m}$ to ensure gradients pass through the communication process. However, since gradient information becomes inaccurate for updated policies, we can only use sampled episodes once, making our method on-policy. In decentralized execution sessions, we simply utilize the shared parameter network to manage greedy actions and one-hot vector messages sampled from Gumbel-Softmax.

## A.2  ICP IMPLEMENTATION WITH RGMCOMM AND VDN

In the pre-training phase of this approach, we used RGMComm, which utilized Regularized Information Maximization loss (Chen et al., 2024) to generate discrete messages. This method represents the latest state-of-the-art explicit communication algorithm in multi-agent reinforcement learning (MARL).

RGMComm itself is based on the MADDPG (Multi-Agent Deep Deterministic Policy Gradient) framework (Lowe et al., 2017). Initially, a full observability centralized actor-critic model (Konda & Tsitsiklis, 1999) is trained to develop a policy with access to global information. This centralized policy serves as a reference to guide the communication strategy for each agent. During this stage, the agents use a centralized critic with a complete view of the environment to assess the action-value functions and optimize the communication strategy. After training this full-observability policy, RGMComm samples different information sets based on local observations. It then clusters these sampled information sets (Harsanyi, 1967) to assign specific messages to each cluster. By doing so, RGMComm derives a discrete communication protocol tailored to each agent's local observations, optimizing their joint policy under partial observability.

In our implementation, instead of using MADDPG, we employed an RDQN (Recurrent Deep Q-Network) (Hausknecht & Stone, 2015) to train the policy with full observability. For example, in environments like Hanabi, the difference between the full observation (global view) and the local observation is minimal, with only additional knowledge of the agent's own hand being considered. Using this full-observation policy, we performed clustering on the information sets, similarly to the RGMComm approach, to obtain the message strategy.

Following the pre-training phase, we proceed to the fine-tuning phase. This phase involves using VDN (Value Decomposition Network) to train the action policy. During this process, the communication strategy is embedded into the scouting action space and is fine-tuned to adapt to the environ-

---

**Algorithm 1:** ICP implementation with DIAL and VDN

---

**Input:** environment **env**, max-train-step, max-episode-length, target-update-rate, $\gamma$, exploration constant $\epsilon$, learning rate $\alpha$, selection of information action space $U_{\text{info}}$

**Output:** Trained model parameters

1 Initialize shared RNN network weights $(\theta_1, \theta_2)$ for each agent $i$ and centralized Mapping Mechanism $\mathcal{P}$, target net weights $(\theta'_1, \theta'_2) \leftarrow (\theta_1, \theta_2)$;

2 **for** *Train step $N = 1, 2, \ldots, $ max-train-step* **do**

3      Reset environment, $o_0 = \textbf{env}.\text{reset}()$;

4      Reset buffer $B$;

5      Initialize empty one-hot messages $m_0$ and target empty one-hot messages $m'_0$, 2 RNNs' hidden states $(h_1, h_2)$ and 2 target RNNs' hidden states $(h'_1, h'_2)$;

6      **for** *time step $t = 1, 2, \ldots, $ max-episode-length* **do**

7          Initialize empty one-hot messages $m_t$;

8          **for** *each agent $i$* **do**

9              Sample message and update hidden state: $m_{\text{sample}}, h_2 = \text{GumbelSoftmax}(\varphi_{\theta_2}(o_{t,i}, m_{t-1,-i}, h_2))$;

10             Sample a random action $u_{\text{random}} \in U_{\text{non-info}}$;

11             Compute greedy action and update hidden state: $u_{\text{greedy}}, h_1 = \arg\max_u(\pi_{\theta_1}(o_{t,i}, m_{t-1,-i}, h_1))$;

12             $\epsilon$-greedy sample action $u_{t,i} \leftarrow \epsilon \cdot u_{\text{random}} + (1 - \epsilon) \cdot u_{\text{greedy}}$;

13             Get action $u_{t,i}$'s q: $q_{t,i} = Q_{\theta_1}(u_{t,i} \mid o_{t,i}, m_{t-1,-i}, h_1)$;

14             **if** $u_{t,i} = \mid U_{\text{non-info}} - 1 \mid$ *(send_info)* **then**

15                 Update action $u_{t,i} \leftarrow u_{info} = \mathcal{P}(m_{\text{sample}}, \bigcap_{i \in N} \tau_{t,i})$;

16                 Update message $m_{t,i} = m_{\text{sample}}$;

17             Sample target message and update hidden state: $m'_{\text{sample}}, h'_2 = \text{GumbelSoftmax}(\varphi_{\theta'_2}(o_{t,i}, m'_{t-1,-i}, h'_2))$;

18             Compute target greedy action's q and update hidden state: $q'_{t,i}, h'_1 = \max_u(\pi_{\theta'_1}(o_{t,i}, m'_{t-1,-i}, h'_1))$;

19             **if** $\arg\max_u(\pi_{\theta'_1}(o_{t,i}, m'_{t-1,-i}, h'_1)) = \mid U_{\text{non-info}} - 1 \mid$ *(send_info)* **then**

20                 Update message $m'_{t,i} = m'_{\text{sample}}$;

21          Step environment, $o_{t+1}, r_t = \textbf{env}.\text{step}(u_t)$;

22          Store $(q_t, q'_t, r_t)$ into buffer $B$;

23      Use samples in buffer to compute sum-q's TD-error: $\delta_t = r_t + \gamma \cdot \sum_i(q'_{t+1,i}) - \sum_i(q_{t,i})$;

24      Update network weights $\theta_1, \theta_2$ using Adam Optimizer with loss $\delta^2_{t,i}$;

25      **if** *Train step $N$* $\mod$ *target-update-rate* $== 0$ **then**

26          Update target network weights $(\theta'_1, \theta'_2) \leftarrow (\theta_1, \theta_2)$;

---

ment. By freezing the RGMComm-generated communication strategy during the initial stages of fine-tuning, we ensure stability and later adjust the embedding to better align with the action policy.

### A.3 IMPLEMENTATION OF DIAL-CHEAT AND ICN-DIAL-DM

DIAL with a Direct Communication Channel (DIAL-Cheat): In this approach, we added a discrete direct channel to the original environment, with the message size matching the size of the scouting action space. This channel allows agents to broadcast information while simultaneously taking actions.

ICP Implemented with the Delayed Map Approach (ICN-DIAL-DM): For this method, we first removed scouting actions from the environment and introduced a discrete direct channel (with the same message size as the scouting action space). Using this modified environment, we pre-trained the agents with DIAL to learn the information strategy. The learned information strategy was then transferred back to the original environment for fine-tuning, where both the information strategy and action policy were optimized.

## B HUMAN CONVENTION IN HANABI

*Hanabi* serves as a compelling platform for studying cooperation and ad-hoc teamwork, particularly within the MARL community, while also being an engaging game in its own right. Beyond its academic appeal, many players have developed highly sophisticated strategies, as detailed in the referenced H-Group conventions [1]. These conventions outline a framework that enables players to achieve high scores by adhering to agreed-upon strategies.

---

[1] For more detailed information on these strategies, please refer to the H-Group's conventions and learning path at https://hanabi.github.io/.

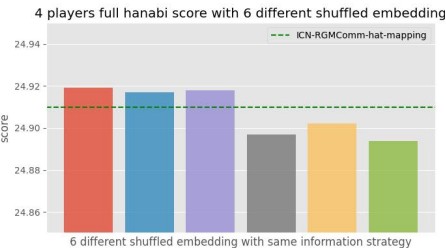

Figure 4: By shuffling the embedding of information into scouting actions, even if the information strategy stays the same, environment information will also change. After fine-tuning, the performance of implementation with the same information strategy but different embedding varies.

Interestingly, some of these strategies embed elements of ToM reasoning into the conventions themselves. For example, techniques like Delayed Play Clues explicitly define the information that certain actions are intended to convey when teammates perform specific moves. Even the Basic and Advanced Strategies, such as misleading bluffs, leverage second-order ToM reasoning to facilitate more effective collaboration.

In a manner similar to ICP, the H-Group conventions construct a stable implicit communication channel by explicitly codifying the ToM reasoning into the actions. When all players follow these conventions, actions serve as reliable carriers of information, ensuring a consistent and effective communication channel within the game.

Compared with H-Group conventions, our ICP framework differs in that the communication strategy itself is learned as part of the training process, rather than being predefined. The communication through informative actions becomes akin to following a *learned convention* where agents develop their own patterns for encoding and transmitting information during the training phase. This allows agents to use scouting or informative actions as explicit communication mechanisms, while regular actions are selected through an online decision-making policy during gameplay. Notably, our method achieves a score of approximately 24.91 in the *Hanabi* game, which significantly surpasses the average human score of around 17 (Kantack, 2021). Thus, ICP offers a dynamic communication framework where the convention of communication emerges from the agents' learning process, enabling more flexible and adaptive strategies depending on the environment and task at hand.

## C  ANALYSIS OF HANABI RESULTS

In Figure 3(b), we present a comparison between our proposed method and SAD, focusing on differences in average scores. While the difference in average score appears minimal in the figure, a detailed analysis provided below reveals that our method achieves a substantial improvement in overall performance.

According to the results reported in the original SAD paper, SAD achieved an average score of 23.81/25 with a win rate of **41.45%**. In contrast, our proposed method achieved an average score of 24.91/25 with a win rate of **91%**. This represents a more than twofold increase in the win rate. The relatively small difference in average scores, despite the significant improvement in win rates, can be attributed to the score ceiling of 25 in Hanabi.

Hanabi imposes inherent constraints on scoring due to its game structure. Certain starting hands, such as those with multiple high-value cards (e.g., 4s and 5s), can make achieving the maximum score of 25 impossible. Considering these limitations, achieving a win rate of 91%—and scores close to 24 in the remaining 9% of games—demonstrates that our method performs near the theoretical maximum.

