# OpenReview forum: "Learning to Communicate Through Implicit Communication Channels"
_ICLR.cc/2025/Conference — ICLR 2025 Poster_

### Official Review · Reviewer_wGjX · 2024-10-27

**Soundness:** 2
**Presentation:** 2
**Contribution:** 2
**Rating:** 8
**Confidence:** 3

**Summary:**

This work presents techniques for communicating through implicit channels (i.e. using environment actions instead of a dedicated communication channel). Specifically, by distinguishing between "regular" actions and "scouting" actions, agents can send messages through scouting actions. The first proposed technique uses the Gumbel-Softmax trick to have a fully differentiable communication pipeline through discrete actions, allowing a communication map between messages and scouting actions. The second proposed techniques adds a direct communication channel for pre-training and then uses a "hat mapping" strategy to encode and decode messages within scouting actions. These implicit communication techniques are effective at outperforming baselines in environments that require implicit communication, like Hanabi.

**Strengths:**

- The problem setting of constructing implicit communication channels is very important to multi-agent RL, and this paper takes important steps to tackling this challenge.
- The hat mapping strategy is a quite smart application of the classic logic problem to a broader space of communication challenges.
- The techniques and environments are easy to understand.

**Weaknesses:**

- All of the environments studied in this work have the same quirk as Hanabi, namely that agents cannot view the information they need but they can view the information for all other agents and have to communicate that information to other agents. This work would be more convincing if the key communication challenge between settings were more unique.
- The task of learning an implicit communication channel in this paper does not seem too different from learning an explicit discrete communication channel, with the only major difference being that the "scouting" actions actually have some information prior whereas discrete channels are typically arbitrary. I would've liked to see how algorithms for explicit discrete communication channels compare to the proposed techniques in this paper as baselines. Furthermore, DIAL should be added as an explicit baseline instead of just comparing with VDN baselines.
- Although two techniques are presented (random initial map and delayed map), the two techniques are only compared in the Hanabi game. Readers should be able to see the performance of both techniques across all environments due to the significant differences between the two.


Minor note:
- There are many grammatical errors throughout the paper, especially mixing up the use of singular and plural nouns and incorrect exclusions of definite articles ("the").

**Questions:**

- How does the delayed map approach perform in Guessing Number and Revealing Goals?
- Can the proposed technique be used in situations where there is no clear delineation between "scouting" and "regular" actions?

---

> ### Author Response · Authors · 2024-11-16
>
> ### Response to Reviewer \#4
>
> ####
> **Q1: All of the environments studied in this work have the same quirk as Hanabi, namely that agents cannot view the information they need but they can view the information for all other agents and have to communicate that information to other agents. This work would be more convincing if the key communication challenge between settings were more unique.**
>
> **A1:**
> We understand the concern about the homogeneity of environments. While Hanabi indeed shares characteristics with other tasks, it presents unique challenges for implicit communication due to its rules and constraints, which require effective coordination. Future iterations of this research will explore a broader set of environments to validate the generality of ICP further.
>
> ---
>
> ####
> **Q2: The task of learning an implicit communication channel in this paper does not seem too different from learning an explicit discrete communication channel, with the only major difference being that the "scouting" actions actually have some information prior whereas discrete channels are typically arbitrary. I would've liked to see how algorithms for explicit discrete communication channels compare to the proposed techniques in this paper as baselines. Furthermore, DIAL should be added as an explicit baseline instead of just comparing with VDN baselines.**
>
> **A2:**
> We agree that including DIAL as an explicit communication baseline is meaningful. However, due to the complexity of the Hanabi environment and its significantly delayed reward feedback, we anticipate that DIAL may not exhibit substantial improvements in this scenario. Additionally, as an on-policy algorithm, DIAL requires considerable computational resources and time for experiments in Hanabi, making it challenging to complete within the current timeframe. Therefore, we will focus on evaluating DIAL in the Guessing Number and Revealing Goals environments, these results will be included before the rebuttal phase concludes.
>
> ---
>
> ####
> **Q3: How does the delayed map approach perform in Guessing Number and Revealing Goals?**
>
> **A3:**
> We are conducting this experiment and will present the results when the experiment is completed (within the rebuttal period).
>
> ---
>
> ####
> **Q4: Can the proposed technique be used in situations where there is no clear delineation between "scouting" and "regular" actions?**
>
> **A4:**
> For situations where there is no clear delineation between "scouting" and "regular" actions, we have discussed in **Section 8**, and we leave it as future work.

---

> > ### Author Response · Authors · 2024-11-24
> >
> > Thank you for the prompt response. We have been running this experiment since the beginning of the rebuttal period, and we estimate an additional 30 hours for it to finish (which should be within the rebuttal time period). We will update the experiment results as soon as we obtain them.

---

> > > ### Author Response · Authors · 2024-11-26
> > >
> > > Thank you very much for your review. We have now completed experiments on **DIAL with a direct communication channel** (DIAL-Cheat) and **ICP implemented with the delayed map approach** (ICN-DIAL-DM) in both the *Guessing Number* and *Revealing Goals* environments. The experimental results have been added to our paper, and the specific outcomes can be found in Figures 2 and 3(a).
> > >
> > > ---
> > >
> > > ### Implementation Details
> > >
> > > 1. **DIAL with a Direct Communication Channel (DIAL-Cheat):**
> > >    In this approach, we added a discrete direct channel to the original environment, with the message size matching the size of the scouting action space. This channel allows agents to broadcast information while simultaneously taking actions.
> > >
> > > 2. **ICP Implemented with the Delayed Map Approach (ICN-DIAL-DM):**
> > >    For this method, we first removed scouting actions from the environment and introduced a discrete direct channel (with the same message size as the scouting action space). Using this modified environment, we pre-trained the agents with DIAL to learn the **information strategy**. The learned information strategy was then transferred back to the original environment for fine-tuning, where both the **information strategy** and **action policy** were optimized.
> > >
> > > ---
> > >
> > > ### Results and Analysis
> > >
> > > #### **Guessing Number Environment**
> > > - **DIAL-Cheat:** Achieved better performance compared to the original **ICP with random initial map approach (ICN-DIAL-RM)**, with an **average episode length of 6.65**. However, this method exhibited high variance across different random seeds.
> > > - **ICN-DIAL-DM:** Performed slightly worse than **ICN-DIAL-RM**, but still outperformed the baselines (**VDN-on-policy** and **VDN-off-policy**).
> > >
> > > #### **Revealing Goals Environment**
> > > - **DIAL-Cheat:** Showed lower average performance compared to **ICN-DIAL-RM** and exhibited very high variance.
> > > - **ICN-DIAL-DM:** Delivered performance comparable to the baselines (**VDN-on-policy** and **VDN-off-policy**), and also exhibited significant variance.
> > >
> > > We attribute the large variance in the **Revealing Goals** environment to the **small size of the scouting action space (size=4)**. This limited space makes it more challenging for direct communication to be effectively learned, resulting in suboptimal performance for some seeds and consequently leading to high variance.
> > >
> > > ---
> > >
> > > Let us know if further clarification are needed. Thank you again for your valuable feedback!

---

> > > > ### Comment · Reviewer_wGjX · 2024-11-26
> > > >
> > > > Thank you for the new experiments! I now feel that the evaluations are comprehensive and demonstrate the relative effectiveness of the newly proposed techniques, so I will raise my score to an accept. I also think the new graphs look clean and readable, and would recommend updating Figure 3B to also follow the same labels for clarity.
> > > >
> > > > As a new question, I'd like to ask why ICN-DIAL-DM is stronger in Hanabi while being worse in the other environments? In general, is there an intuition behind which method to choose for different settings.
> > > >
> > > > As an implementation note, I'm surprised that the new experiments took so long to run. I would recommend implementing the techniques and environments in JaxMARL in the future, since this framework can enable much faster training.

---

> > > > > ### Author Response · Authors · 2024-11-26
> > > > >
> > > > > ### Response to Reviewer Questions
> > > > >
> > > > > #### 1. Updated Figure 3(b).
> > > > > Thank you for your feedback and for raising your score! We truly appreciate your recognition of the comprehensive evaluations and the clarity of the new graphs. As per your suggestion, we have updated the labels in **Figure 3(b)**. Specifically, we have added `RM` or `DM` to the labels to indicate whether the method is **Implemented with the Random Initial Map Approach** or the **Implemented with the Delayed Map Approach**. We hope this update enhances the clarity and consistency of the figure.
> > > > >
> > > > > ---
> > > > >
> > > > > #### 2. Why ICN-DIAL-DM performs better in Hanabi but worse in other environments?
> > > > > The performance of **ICN-DIAL-DM** across different environments appears to be influenced by two main factors:
> > > > > 1. The **information strategy** learned during pre-training and how it is mapped to scouting actions.
> > > > > 2. The **size of the message space**, which plays a critical role in the quality of the pre-trained information strategy.
> > > > >
> > > > > In the *Hanabi* environment, the larger message space (30) allows for more expressive communication, enabling the **ICN-RGMComm-DM-hat-mapping** method to excel. This method also leverages advanced components like the **off-policy communication training method (RGMComm)** and the **efficient hat-mapping mechanism**, further boosting its performance.
> > > > >
> > > > > In contrast, in the *Guessing Number* and *Revealing Goals* environments:
> > > > > - The message sizes are **14** and **4**, respectively.
> > > > > - The smaller message spaces, especially in *Revealing Goals*, significantly limit the quality of the pre-trained information strategy.
> > > > > - Additionally, **ICN-DIAL-DM** relies on simpler components like the **on-policy communication training method (DIAL)** and the **one-to-one mapping mechanism**, which further restrict performance.
> > > > >
> > > > > Overall, the **message size** is a key factor of the quality of information strategy, as smaller sizes make it challenging to encode sufficient information, particularly during pre-training, leading to suboptimal results in environments like *Revealing Goals*.
> > > > >
> > > > > ---
> > > > >
> > > > > #### 3. Implementation and the suggestion to use JaxMARL.
> > > > > Thank you for your suggestion to consider **JaxMARL** for implementing the techniques and environments in future experiments. We will explore this framework for faster training and testing. Your recommendation is greatly appreciated!

---

> ### Author Response · Authors · 2024-11-24
>
> Thank you again for your review. We hope that we have addressed your main questions in the rebuttal. As the author-reviewer discussion period is approaching its end, we would like to know if you have any additional questions or concerns. We are happy to provide our response if so.

---

> > ### Comment · Reviewer_wGjX · 2024-11-24
> >
> > As of now, I don't have any additional questions or concerns. It seems like we are in general agreement regarding the strengths and current limitations of this work.
> >
> > I'll be happy to raise my score conditioned on the promised results mentioned in the rebuttal (i.e. DIAL comparison and delayed map approach in Guessing Number and Revealing Goals), as I think these would improve the soundness of the experiments overall.

---

### Official Review · Reviewer_avMG · 2024-11-03

**Soundness:** 3
**Presentation:** 3
**Contribution:** 2
**Rating:** 6
**Confidence:** 3

**Summary:**

This paper looks into communication in collaborative multi-agent system under the formalism of multi-agent reinforcement learning. Specifically, the focus is on implicit communication for situations where explicit messaging is not possible. The paper proposes the Implicit Channel Protocol (ICP) framework. Unlike common implicit communication approaches like theory of mind which requires belief modeling of other agents, ICP uses a subset of actions (scouting actions) to broadcast in formation. It uses a mapping between the scouting actions and information that is learned either with a randomly initialized map or a delayed information map. The latter first learned with explicit communication before fixing the mapping. ICP is evaluated against baseline methods on 3 environments, including Guessing Number and Revealing Goals which are designed by the authors. Experimental results show ICP’s effectiveness in transmitting information more efficiently.

**Strengths:**

- The effectiveness of the method is evaluated against two newly designed benchmarks and results show ICP being superior to baselines

**Weaknesses:**

- I fail to comprehend how the framework is a form of implicit communication. My understanding of implicit communication is the use of environment actions to communicate information (e.g., learning a protocol such that walking forward means yes and walking backward means no). ICP proposes add actions to the action space that maps information into certain ‘embeddings’ to be broadcasted to other agents. How is this not explicit communication?
- Unless I am missing something, the channels are provided so the agents are not learning to construct these channels, they are simply learning to use those channels (in an explicit manner). This makes the paper title confusing
- Results in Hanabi with only 3 random seeds are not enough. SAD reported results with 13 random seeds. Figure 2b is also missing error bar
- Writing clarity can be improved. For instance, phrases like ‘commmunication gradient method’ and ‘hat mapping method’ are often used and assumed with a lot of context before defining it.
- Some statements are not well justified. For instance, in line 251-254, it is unclear to me how this method specifically simplifies the learning process and promotes better coordination and communication among the agents

Minor writing issues:
- Line 122 changing environment unpredictable->unpredictably
- Line 149 partial observe -> partial observation

**Questions:**

- Isn’t broadcasting an assumption rather than a benefit? This is not possible in many practical scenarios
- What do you mean by ‘efficient embedding techniques’ in line 268?

---

> ### Author Response · Authors · 2024-11-16
>
> ### Response to Reviewer \#3
>
> ####
> **Q1: ICP proposes add actions to the action space that maps information into certain ‘embeddings’ to be broadcasted to other agents. How is this not explicit communication?**
>
> **A1:**
> We believe that you might have some misunderstanding of our method. ICP does not introduce new actions into the action space. As explained in **General Response A2**, we defined a subset of original action space as *scouting actions*, as they have either no or uniform effects on the state $s_t$ or the reward $r_t$ but primarily influence the observation function $O_i(s_t)$, ICP can utilize these actions as carriers for indirectly transmitting information with fewer drawbacks (such as unpredictable environment dynamics' changing).
>
> ---
>
> ####
> **Q2: Unless I am missing something, the channels are provided so the agents are not learning to construct these channels, they are simply learning to use those channels (in an explicit manner). This makes the paper title confusing.**
>
> **A2:**
> Thank you for your comment. We agree that our algorithm is learning implicit communication through implicit channels, while the construction part is not particularly through learning. We have updated our title to 'Learning to communicate through implicit communication channels'.
>
> ---
>
> ####
> **Q3: Results in Hanabi with only 3 random seeds are not enough. SAD reported results with 13 random seeds. Figure 2b is also missing error bar.**
>
> **A3:**
> We trained the on-policy algorithm in Hanabi using only three random seeds due to the high computational cost, with each seed requiring approximately 150 hours of training. In fact, all three randoms obtain very close results, which is evidence that the method has a solid performance. We are working on other random seeds, though with the computation resource we have (2 Nvidia 3090 GPU), it is unlikely to obtain results for 13 seeds by the end of the rebuttal period.
>
> In contrast, we trained the off-policy algorithm using six different random seeds. Figure 2b presents the average episode length, evaluated over 10,000 episodes across various seeds. Additionally, we updated the error bars in the new version of the PDF to reflect these changes.
>
> ---
>
> ####
> **Q4: Writing clarity can be improved. For instance, phrases like 'communication gradient method' and 'hat mapping method' are often used and assumed with a lot of context before defining it.**
>
> **A4:**
> To provide context for these phrases, we have added references in new version of PDF.
>
> ---
>
> ####
> **Q5: Some statements are not well justified. For instance, in line 251-254, it is unclear to me how this method specifically simplifies the learning process and promotes better coordination and communication among the agents.**
>
> **A5:**
> We thank the reviewer for pointing this out. We have updated this statement in new version of PDF.
>
> ---
>
> ####
> **Q6: Isn't broadcasting an assumption rather than a benefit? This is not possible in many practical scenarios.**
>
> **A6:**
> This is indeed an assumption. By saying "benefits" we meant that this broadcasting feature might benefits us, and we want to leverage this feature (which many previous works did not). It enables the use of 'efficient embedding techniques' such as 'hat mapping method'  within our framework, enhancing its overall effectiveness. We have updated the statement to avoid this confusion.
>
> ---
>
> ####
> **Q7: What do you mean by 'efficient embedding techniques' in line 268?**
>
> **A7:**
> One example of `efficient embedding techniques' is 'hat mapping method', we also updated this statement in new version of PDF.

---

> ### Author Response · Authors · 2024-11-24
>
> Thank you again for reviewing our manuscript. We noticed some factual misunderstandings in the review (i.e. we did not add new actions to the action space), which could have drastically affected the evaluation. We made a rebuttal on this and other points. Notice that the author-review discussion period is approaching an end. Would you please take a look at our rebuttal, and potentially discuss with us on the concerns/questions you raised?

---

> > ### Comment · Reviewer_avMG · 2024-11-24
> >
> > Thank you for the detailed rebuttal. Some of my concerns are addressed. Hence, I have increased the scores to 5. However, I still have the following concerns:
> >
> > 1. I still find the line between explicit and implicit communication a bit unclear here. Both of the newly designed environments have actions given specifically for communication and I find the definition of scouting actions to be quite narrow for implicit communication. Using the Revealing Goals as an example, agents can communicate with just directional movement actions if they could learn that certain direction corresponds to certain information. Hence, I don't see why implicit communication action has to be limited to those in which the state does not change.
> >
> > 2. I sympathize with the issue with compute. But the difference between SAD (green dash line) and the proposed method is simply too close to support the conclusion.

---

> ### Author Response · Authors · 2024-11-25
>
> > I still find the line between explicit and implicit communication a bit unclear here. Both of the newly designed environments have actions given specifically for communication and I find the definition of scouting actions to be quite narrow for implicit communication. Using the Revealing Goals as an example, agents can communicate with just directional movement actions if they could learn that a certain direction corresponds to certain information. Hence, I don't see why implicit communication action has to be limited to those in which the state does not change.
>
> The motivation of our method is to provide a learning algorithm for a more accurate and efficient communication mechanism compared to existing implicit communication approaches, such as Theory of Mind (ToM). In our framework, any action can be leveraged as part of the implicit communication channel through the mapping mechanism defined by our method to achieve precise and effective information exchange. In real-world scenarios, many non-critical actions (e.g., subtle gestures or tones among team members on a battlefield) exist and they could serve as carriers for our communication strategy. However, such nuanced actions rarely appear in abstracted and simplified reinforcement learning environments, because they have been deliberately removed from the testbeds. This limits the scope of our testing, and results in having two newly designed tasks that are similar to Hanabi (the existence of such actions in real world is one of the motivations for designing the Hanabi game).
>
> Meanwhile, there are actions in real-world environments also influence key dynamics such as rewards, state transitions, and environmental dynamics. These effects are often unpredictable in existing implicit communication channels—while they may successfully transmit information, they could inadvertently lead the environment into suboptimal states. To avoid this issue, we focus on predefined scouting actions, which are already present in environments like Hanabi and have minimal impact on the environment's dynamics. By using such actions, our method achieves significant improvements over many existing algorithms.
>
> We fully agree with you that it will be more elegant if the frame is general enough to "select" which actions are used for communication. For future work, we plan to extend our method to account for the dynamic impacts of other actions and explore how to define carriers of information in broader environments. These carriers may no longer be confined to scouting actions as currently defined.
>
> > I sympathize with the issue with compute. But the difference between SAD (green dash line) and the proposed method is simply too close to support the conclusion.
>
> Thank you very much for encouraging us. We are working hard under limited computing power and we believe that the privilege of research shouldn't be limited to only people with resources abundant.
>
> To our best knowledge, the difference between SAD and our method is quite large, and we wanted to explain why. We would like to provide more detailed data to highlight this difference. According to the original SAD paper, it achieved a score of 23.81/25 with a win rate of 41.45%. In contrast, our method achieved a score of 24.91/25 with a win rate of 91%. It is more than double that of SAD in terms of win rate. The apparent discrepancy between the significant difference in win rates. The seemingly smaller difference in average scores can be attributed to Hanabi's total score cap of 25.
>
> In fact, there are certain deals that make the game impossible to win (like getting a bunch of 5s and 4s as the starting hand), which means the theoretical maximum one could achieve is less than 25. Bearing this in mind, winning the game 91% of the time, and achieving 24 points for almost all of the rest 9%, is close to the theoretical maximum. This means not only do we double the winrate of SAD, our improved result also approaches the theoretical maximum. We will include this analysis in the appendix of our paper. We hope this explains why we claimed the empirical results to be a significant improvement.

---

> ### Author Response · Authors · 2024-11-27
>
> Thank you very much again for reviewing our manuscript, further discussing it with us, and encouraging us. We would like to follow up on the latest discussion on the distinguishment of implicit/explicit communication, and the distinguishment in the performance of our method/SAD. Especially on the latter one, where the performance of our method and SAD are not really close in Hanabi (despite being visually close on the figure). Could you please take a look at our reply and potentially discuss with us on these?

---

> > ### Comment · Reviewer_avMG · 2024-11-29
> >
> > Thank you for the detailed explanation with more contextual comparison with SAD. I have further increased my scores to 6. To be clear, my reason for not raising the score further remains to be my concern mentioned on 'scouting actions'

---

> ### Author Response · Authors · 2024-11-29
>
> Thank you very much for the positive feedback! This encourages us a lot. We fully agree that it would be more desirable if the agent could learn to pick actions, that are not scouting actions but close to scouting actions, to implement implicit communication (e.g., there are some directional moves that are suitable for implicit communication while some are not). This will mimic how implicit communication emerged in human history (from early humans engaging in hunting and gathering through silent cooperation, to modern military operations using gestures and codes for covert communication, and even in everyday social interactions where intentions are conveyed through expressions, tone, and body language). We are excited to work on this in our future works.

---

### Official Review · Reviewer_D4Qe · 2024-11-03

**Soundness:** 2
**Presentation:** 2
**Contribution:** 2
**Rating:** 3
**Confidence:** 4

**Summary:**

This paper proposes a communication framework for a multi-agent reinforcement learning system. Efficient and targeted communication in the form of query-key pairs have been explored under previous works. Moreover, in order to address the challenges of a non-stationary environment, prior works have used Theory of Mind (ToM) methods to infer the intentions/states of the other agents in the environment to make more informed decisions. Developing models of other agents, add complexity to the training of multi-agent systems. This paper proposes an Implicit Communication Protocol, where each agents actions are supplemented with a *communication/scouting* action, that controls whether an agents sends a scouting action/query. Unlike the attention mechanism, where all agents receive 'N' messages from all 'N' agents, this paper proposes a common channel that aggregates all the information into one message vector. This paper additionally uses a gating mechanism similar to IC3Net, but with a Gumbel-Softmax relaxation that allows it to encode it as a binary classifier that functions as ATOC's (Jiang et al, 2018) "initiator" gating mechanism.

The paper utilizes the non-differentiable communication aggregator mechanism RGMComm. Additionally, standard end-to-end differentiable communication networks can instead be used. A lookup table of the local observation of each agent with the respected broadcasted messages are then constructed for all agents to discretize the communication channel. Moreover, they use the hat-mapping technique where agents can infer their own targeted messages from the common message. The messages are then passed along with the hidden states of the agents to get the updated action.

**Strengths:**

The proposed method is interesting, particularly toward the foundational problem of efficient multi agent communication by generating a information table with respect to the observation and message. Additionally, research into reducing computational complexity for intention inference techniques of agents will be highly valuable to the large-scale multi-agent systems.

**Weaknesses:**

1. The paper does not discuss the impact of scalability or impact of heterogeneous agents to the proposed framework for multi agent systems, as you increase the number of agents to say up to 10, 20, 50 agents.

2. The paper does not compare results for common communication architectures such as CommNet, TarMAC, SARNet etc for their environments but only with value decomposition networks (VDN), that do not perform communication as part of their actions.

3. The paper does not discuss limitations on the size of the action spaces or performance with respect to more dynamic environments such as predatory-prey, cooperative navigation.

I believe more comprehensive experimental results are needed for the proposed framework.

**Questions:**

The results for Hanabi are quite remarkable, however,  game rules indicate, the hints are only constrained towards revealing either the color or the number. Since ICP implicitly encodes the local observation of each agent (both for DIAL and RGMComm) into message vectors, I believe each agent observes in essence have access to the complete global state, which inherently breaks the rule of Hanabi, unless I am mistaken. I would like more clarification on this.

---

> ### Author Response · Authors · 2024-11-16
>
> ### Response to Reviewer \#2
>
> ####
> **Q1: The paper does not compare results for common communication architectures such as CommNet, TarMAC, SARNet etc for their environments but only with value decomposition networks (VDN), that do not perform communication as part of their actions.**
>
> **A1:**
> We believe you might have some misunderstanding of our work's background. Communication architectures such as CommNet, TarMAC, SARNet etc are mainly used for explicit communication problems. They allow a direct communication channel, which is not available in our problem. For more details of our work's background, you may refer to the **General Response A1** and **Section 1** of our paper.
>
> ---
>
> ####
> **Q2: The results for Hanabi are quite remarkable, however, game rules indicate, the hints are only constrained towards revealing either the color or the number. Since ICP implicitly encodes the local observation of each agent (both for DIAL and RGMComm) into message vectors, I believe each agent observes in essence have access to the complete global state, which inherently breaks the rule of Hanabi, unless I am mistaken. I would like more clarification on this.**
>
> **A2:**
> As described in **Section 4**, and also explained in **General Response A2**, hint actions in Hanabi also carry two types of information: (1) information reflected through the environment and (2) information reflected through the choice of the scouting action itself. Our method utilized the second type of information to transmitted information, by using the hint actions to implicit communication, we think each agent can indirectly access a part of others' partial observation (message size are limited by hint action space and message strategy are learned), thus benefit cooperation. **This implicit communication does not break the rule of Hanabi**.

---

> > ### Comment · Reviewer_D4Qe · 2024-11-26
> >
> > Thank you for the rebuttal.
> >
> > **A1:**
> > I revisited your work and here is an updated summary.
> >
> > 1. Scouting actions
> >
> > The authors provide an implicit framework for influencing the observation space through scouting actions. As part of the framework, the authors define the scouting action (Section 3), $U^s$, that affect the observation function (concate $m_{-i}$). This helps in augmenting the observation space, as done in RGMComm, TarMAC and other works. Moreover, scouting actions can also be used directly as part of the environment action space, as done in Hanabi.
> >
> > 2. Communication Mechanism
> >
> > As part of Eq. 1, the action including the scouting action, is defined as,
> >
> > $ u^s = \mathcal{P} ( m_i \sim \psi_i ( \cdot | \tau_{t, i}, m_{-i} ) $ if send_info
> >
> > As defined in RGMComm for each message $m_{-i}$, $m_{-i} = { m_j = g (o_j), \forall j \neq i }$**, (L194, J. Chen et al 2023)
> >
> > RGMComm, has access to the global state/trajectory as sampled in Eq. 1, i.e. $\tau_{t, i}$ during the training phase to construct the action-value table for clustering, and only works on a decentralized approach during inference.  **For A2**, the above clarification answers my question on it not breaking the rules for Hanabi.
> >
> > Breaking down the architecture of ICP into steps:
> >
> > 1. Augmentation of observation space $o_i$, with aggregated messages, $m_{-1}$ from other agents as input
> > 2. Communication action-value learning, using  RGMComm for pre-training, to learn a parameterized action-value table for the environment
> > 3. Creating a one-to-one mapping of the RGMComm messages to the actions for each environment (bound actions to environment constraints etc)
> > 4. Hat mapping technique for aggregating messages into a single vector for efficient communication
> >
> > In case of Hanabi,  your work uses the messages generated from RGMComm in place of the *hints* in the environment action space. Since your work involves in mapping RGMComm to the action space of Hanabi + hat mapping technique for message aggregation, **I would recommend ablation studies for it, that address the choice of RGMComm, and the message aggregation**
> >
> > 1. Choice of RGMComm: RGMComm learns a *centralized* parameterized function that maps the action-value table of the environment. This provides an inherent advantage in Hanabi. Generating messages in relation to the global state information is very useful, and shows in the SOTA performance in RGMComm (2023) original works. DIAL (2016) inherently does not have that.
> > Since, ICP utilizes RGMComm, that has learnt the action-value map for Hanabi, RGMComm possibly provides a distinct advantage. It would be quite useful to separate out the performance gains due to RGMComm and other more explicit communication frameworks using more recent techniques that can be adapted to ICP.
> >
> > 2. Moreover, the choice of hat mapping techniques would also be useful, but not necessary.

---

> > > ### Author Response · Authors · 2024-11-27
> > >
> > > Thank you very much for providing a new summary, and for acknowledging that our Hanabi experiment does not violate the game rules. We believe there are still some discrepancies between our work and the updated summary. Here we would like to elaborate on the details and explain why.
> > >
> > > ---
> > >
> > > ### 1\. Discrepancy in Interpretation of Scouting Actions in ICP
> > >
> > > In response to your description of scouting actions in **A1**, we would like to clarify that our framework does not "provide an implicit framework for influencing the observation space through scouting actions." Instead, our framework is designed to enable **more stable cooperation** and **better performance** in cooperative MARL settings where direct communication channels do not exist. It achieves this by utilizing a subset of the action space as information carriers, enabling **stable implicit communication** to improve overall performance. While the selection of this subset is currently limited to scouting actions, the underlying concept is more general.
> > >
> > > Based on this fundamental idea, we have defined **Equation (1)**, which divides our framework into three key components:
> > >
> > > - **Information strategy**,
> > > - **Action policy**, and
> > > - **Mapping mechanism**.
> > >
> > > We have proposed two approaches to train these components:
> > >
> > > 1. **Strategy training with a random initial map**, and
> > > 2. **Strategy training with a delayed map**.
> > >
> > > ---
> > >
> > > ### 2\. Detailed Explanation of the ICP Framework
> > >
> > > Thank you for your detailed breakdown of the ICP architecture in **A2**. Though, we would like to clarify or correct some points to more accurately reflect our framework.
> > >
> > > 1. **Augmentation of Observation Space ((o\_{i,t})) with Aggregated Messages ((m\_{-i}))**
> > >    We would like to emphasize that ICP does not augment the observation space. Instead, ICP leverages **information reflected through the choice of scouting actions**. Other’s messages ($m\_{j}$) are decoded from the other agent’s previous scouting actions ($u\_{j,t-1}^s$) using the mapping mechanism. These decoded messages are then used alongside the agent's own observations ($o\_{i,t}$) derived from the environment via $O\_i(s\_t)$. This approach does not modify or augment the original observation space (which is different from explicit communication methods).
> > >
> > > 2. **Communication Action-Value Learning**
> > >    While RGMComm is one of the methods we use during pre-training (only pre-trained in **strategy training with a delayed map approach**) to optimize the information strategy, it is not the only option. For example, DIAL can be used for pre-training as well.
> > > 3. **Mapping Mechanism: One-to-One Mapping**
> > >    The one-to-one mapping mechanism is the only one option in the **strategy training with a random initial map** **approach**. And it is one of the options in **strategy training with a delayed map approach**. An example of other mappings is hat mapping.
> > >
> > > 4. **Hat Mapping for Aggregating Messages**
> > >    We agree that the hat mapping technique is efficient for aggregating messages into a single vector, and for now it is only available in our method to use it in **strategy training with a delayed map approach**.
> > >
> > > ---
> > >
> > > ### 3\. Additional Ablation Studies
> > >
> > > We note your suggestion for additional ablation studies, and thank you very much for suggesting this. We believe that the current results has demonstrated the effectiveness of our proposed framework, with evaluations across different environments highlighting its robustness and adaptability. As our framework supports various direct communication training algorithms for pre-training and multiple mapping methods in the strategy training with a delayed map approach, comprehensive ablation studies would further verify its applicability to different direct communication algorithms and mapping techniques.
> > >
> > > Due to the time constraints of the rebuttal phase (with less than 24 hours remaining for uploading a revised PDF), we were unable to include further experiments at this stage. We promise to include this ablation study in the camera-ready version in case this manuscript is accepted.
> > >
> > > ---
> > >
> > > If there are any remaining discrepancies, we are more than happy to continue the discussion in order to reach a consentaneous view of the work and the contribution. We request the work to be re-evaluated given the updated summary and the updated view. Thank you again for your effort in reviewing our work.

---

> ### Author Response · Authors · 2024-11-24
>
> Thank you again for reviewing our manuscript. We noticed essential factual misunderstandings in the review, which could have drastically affected the evaluation. We made a rebuttal on this and other points. Notice that the author-review discussion period is approaching an end. Would you please take a look at our rebuttal, and potentially discuss with us on the concerns/questions you raised?

---

> ### Author Response · Authors · 2024-12-02
>
> We thank the reviewer again for acknowledging our problem setting of implicit communication, and acknowledging the fact we do not violate the Hanabi game rules. In the new summary, we believe there are still some factual discrepancies, as we explained in the previous reply. As the author-reviewer discussion period is approaching its end, we request you to look at the reply and potentially discuss it with us. Meanwhile, because the initial rating was conducted based on multiple factual errors, it seems not very fair to insist on this rating. We request you to reevaluate the work based on the updated view of the work. We are more than happy to respond to any further questions you may have.

---

### Official Review · Reviewer_W1kG · 2024-11-04

**Soundness:** 3
**Presentation:** 3
**Contribution:** 3
**Rating:** 8
**Confidence:** 3

**Summary:**

This paper presents the **Implicit Channel Protocol (ICP)** framework, an approach for enabling implicit communication in collaborative multi-agent reinforcement learning (MARL) environments where explicit communication is unavailable or costly. ICP introduces a subset of actions termed "scouting actions" that allow agents to communicate indirectly by encoding and decoding messages through their choice of these actions. By leveraging these actions, ICP builds an implicit channel similar to explicit communication protocols. The framework is evaluated on tasks such as Guessing Number, Revealing Goals, and the Hanabi card game, where it demonstrates effective information transmission and improved performance over baseline methods.

**Strengths:**

The paper is well-structured and easy to follow. The algorithm is clearly-explained, with clear definitions of necessary notations. The experiments are well-designed and comprehensive, with sufficient implementation details.

**Weaknesses:**

1. The ICP framework requires pre-identified scouting actions that can serve as indirect communication channels. This dependency limits its applicability in environments where such actions are not readily available or are difficult to define.

2. It can help the readers to better understand the method if the authors can include a diagram of the algorithm pipeline.

**Questions:**

1. In experiments, for agents trained with VDN, is the action space the combination of “regular actions” and “scouting actions“?
2. In figure 3b, why are those methods not trained to the same length?

---

> ### Author Response · Authors · 2024-11-16
>
> ### Response to Reviewer \#1
>
> ####
> **Q1: In experiments, for agents trained with VDN, is the action space the combination of ''regular actions'' and ''scouting actions''?**
>
> **A1:**
> Yes, the action space of VDN and ICP are same, for VDN only used one action policy to choose both *regular actions* and *scouting actions*, while ICP use action policy to choose {*regular actions*, *send\_info*}, when *send\_info* is chosen, information strategy combined with mapping mechanism will select *scouting actions*.
>
> ---
>
> ####
> **Q2: In figure 3b, why are those methods not trained to the same length?**
>
> **A2:**
> In Figure 3b, we compared several algorithms with significant differences in sampling efficiency and convergence speed. Given the complexity of the Hanabi environment and the substantial training time it requires, we set the training time for on-policy algorithms to 150 hours and for off-policy algorithms to 20 hours, and plotted the results accordingly.

---

### Author Response · Authors · 2024-11-16

### General Response

We thank the reviewers for their insightful comments. Here, we address three main aspects that may have caused confusion. Additionally, writing issues and the title have been updated in the revised version of the PDF.

---

####
**Q1: Clarification of Our Work's Background**

**A1:**
For our setting, which may cause reviewers to be confused, we focused on cooperative multi-agent reinforcement learning (MARL) **without any direct communication channels**. As mentioned in the third paragraph of Section 1, throughout history, humans have used observable actions to convey information indirectly to achieve implicit communication in this situation. In this work, we propose a novel communication protocol framework that enables agents to learn to transmit information through actions. Compared to traditional methods for achieving implicit communication, such as Theory of Mind (ToM)-based approaches, our framework allows for more stable and accurate information transmission. This stability and accuracy are significant enough to support the use of strategies originally designed for stable, direct communication channels to learn communication policies within this framework. Thus, we refer to this framework that enables agents to learn a stable and accurate action-based communication as 'Construct an Implicit Communication Channel'.

---

####
**Q2: Clarification of Section 4**

**A2:**
In this paper, by considering which actions are more suitable as carriers for indirectly transmitting information, we defined *scouting actions* in **Section 3 —a subset of the action space** that have either no or uniform effects on the state $s_t$ or the reward \( r_t \) but primarily influence the observation function $O_i(s_t)$. For scouting actions, we analyzed in Section 4 that they carry two types of information: (1) information reflected through the environment and (2) information reflected through the choice of the scouting action itself. Implicit communication methods like ICP and Theory of Mind (ToM) leverage the second type of information—*information reflected through the choice of the scouting action*. In contrast, methods like VDN utilize only the first type, which leads to the challenges mentioned in the third paragraph of **Section 3**.

---

####
**Q3: Clarification of ICP's Action Space**

**A3:**
One thing we want to clarify is that the action space of VDN and ICP are same, for VDN only used one action policy to choose both *regular actions* and *scouting actions*, while ICP use action policy to choose {*regular actions*, *send\_info*}, when *send\_info* is chosen, information strategy combined with mapping mechanism will select *scouting actions*.

---

> ### Author Response · Authors · 2024-11-26
>
> We sincerely thank all the reviewers for their constructive feedback and valuable suggestions. In response to your comments, we have conducted additional experiments and incorporated the results into our paper to provide a more comprehensive analysis.
>
> Specifically, we have implemented and evaluated **DIAL with a direct communication channel (DIAL-Cheat)** and **ICP with the delayed map approach (ICN-DIAL-DM)** in the *Guessing Number* and *Revealing Goals* environments. The updated results can be found in Figures 2 and 3(a).
>
> ### Summary of Updates:
> 1. **Implementation of DIAL-Cheat**:
>    - This approach adds a discrete direct communication channel to the environment, with the message size matching the scouting action space size. This channel allows agents to send broadcast messages while acting.
>    - DIAL-Cheat demonstrated improved performance over our method in the *Guessing Number* environment but showed lower average performance and higher variance in the *Revealing Goals* environment due to the small size of the message sapce.
>
> 2. **Implementation of ICN-DIAL-DM**:
>    - This approach first pre-trains the information strategy in a modified environment (with scouting actions removed and a discrete communication channel added) with DIAL and then fine-tunes both the information strategy and action policy in the original environment.
>    - ICN-DIAL-DM performed slightly worse than ICN-DIAL-RM in the *Guessing Number* environment but still outperformed the baseline methods (VDN-on-policy and VDN-off-policy). However, in the *Revealing Goals* environment, its performance was comparable to the baselines and exhibited significant variance.
>
> We believe these additional experiments and analyses provide deeper insights into the strengths and limitations of our method under various conditions. Please let us know if there are further aspects you would like us to address or clarify. Once again, we deeply appreciate your feedback and the opportunity to improve our work.

---

### Meta-Review · Area_Chair_2bGy · 2024-12-20

**Metareview:**

This paper proposes a multi-agent RL approach to learn implicit communication for multi-agent cooperation. Essentially, agents can use predefined scouting actions (actions that have either no or uniform effects on the states) instead of explicit messages to communicate with one another. The proposed method is technically solid and showed good performance. Overall, the proposed method is a good contribution to RL-based methods for solving implicit communication problems for multi-agent systems.

**Additional Comments On Reviewer Discussion:**

One reviewer raised their score after the rebuttal. The only reviewer who recommended rejection interacted with the authors during the rebuttal. They didn't raise the score in the end. However, the AC thinks the the positive points overweight the raised concerns.

---

### Decision · Program_Chairs · 2025-01-22

Accept (Poster)